# A Case of Tracheal Stenosis as an Isolated Form of Immunoproliferative Hyper-IgG4 Disease in a 17-Year-Old Girl

**DOI:** 10.3390/children8070589

**Published:** 2021-07-12

**Authors:** Natalia Gabrovska, Svetlana Velizarova, Albena Spasova, Dimitar Kostadinov, Nikolay Yanev, Hristo Shivachev, Edmond Rangelov, Yanko Pahnev, Zdravka Antonova, Nikola Kartulev, Ivan Terziev, Kaloyan Gabrovski

**Affiliations:** 1Department of Pulmonary Diseases, Multiprofile Hospital for Active Treatment of Pulmonary Diseases “St. Sofia”, Medical University–Sofia, 1431 Sofia, Bulgaria; sv_velizarova@abv.bg (S.V.); doc_spasova@abv.bg (A.S.); dimko@mail.bg (D.K.); dr.nikolay.yanev@gmail.com (N.Y.); 2Department of Pediatric Thoracic Surgery, Pediatric Surgery Clinic, University Multiprofile Hospital for Active Treatment and Emergency Medicine “N. I. Pirogov”, 1606 Sofia, Bulgaria; hshivachev@gmail.com (H.S.); edmondrangelov@abv.bg (E.R.); yanko.pahnev@gmail.com (Y.P.); zantonova1978@abv.bg (Z.A.); n.kartulev@gmail.com (N.K.); 3Department of Pathology, University Hospital ‘’Tsaritsa Uoanna–ISUL’’, 1527 Sofia, Bulgaria; titia@abv.bg; 4Department of Neurosurgery, University Hospital “St. Ivan Rilski”, Medical University–Sofia, 1431 Sofia, Bulgaria; k_gabrovski@abv.bg

**Keywords:** tracheal stenosis, children, IgG4 immunoproliferative disease

## Abstract

Immunoglobulin G4-related disease (IgG4-RD) is a lymphoproliferative disease which is described almost exclusively in adults. There are only a few pediatric patients who have been observed with this disorder. Here, we describe a rare case of IgG4-RD in a 17-year-old girl with a single manifestation—tracheal stenosis without previous intubation or other inciting event. She had mixed dyspnea and noisy and weakened breathing. Immunoproliferative hyper-IgG4 disease was diagnosed, based on elevated serum IgG4 and histological findings. Until now we have chosen to treat the girl only with corticosteroids with a good response so far. The general condition as well as the respiratory function are regularly monitored. The tracheal involvement of IgG4-RD is uncommon. Nonetheless, it is a manifestation that should be included in the differential diagnosis of tracheal stenosis.

## 1. Introduction

Immunoglobulin G4-related disease (IgG4-RD) is a lymphoproliferative disorder which is observed almost exclusively in adults. There are very few pediatric patients who have been described with the disease [1,2].

It is an immune-mediated condition that can affect almost any organ in the body. Common presentations are major salivary and lacrimal gland enlargement, orbital disease, autoimmune pancreatitis (the first described involved organ) and tubulointerstitial nephritis [3,4]. It is characterized by a lymphoplasmacytic infiltrate composed of IgG4^+^ plasma cells, exuberant fibrosis and mild to moderate eosinophilia [5,6].

Usually, the most common cause of acquired tracheal stenosis is a previous intubation, tracheotomy or tracheostomy. Here, we describe a rare case of IgG4-RD in an adolescent girl with a single manifestation—tracheal stenosis without previous intubation or other inciting event.

## 2. Case Presentation

We present a case of 17-year-old girl diagnosed with bronchial asthma and allergic rhinitis at the age of 3, hospitalized many times due to complaints of fatigue and persistent drug-resistant dyspnea—controlled therapy with inhaled corticosteroids was done with unsatisfactory results, the girl was under asthma therapy for 12 years, with no significant change in the presenting complaints. The patient has never been intubated, she had no other diseases in the past.

At the age of 15 the medical examination showed impaired general condition, mixed dyspnea and noisy and weakened breathing. Physical examination was otherwise unimpressive.

The laboratory tests showed peripheral hypereosinophilia, elevated serum IgG levels—18.20 g/L (reference range 5.4–16.1 g/L), elevated serum IgG4 levels—8.25 g/L (reference range 0.23–1.11 g/L), IgG4/IgG ratio—0.45. Spirometry showed: total lung capacity (TLC) 99.2%; forced vital capacity (FVC) 77.8%; forced expiratory volume (FEV1) 83.8%; FEV1/FVC 88.71%; low positive bronchial dilation test—FEV1 fifteen minutes after salbutamol inhalation—93.9%. Chest computed tomography (CT) scan did not show any pathological findings in the lungs. Fibrobronchoscopy was performed—fibrous stenosis was found at the level of the lower edge of the thyroid cartilage ring with an infiltration area (Figure 1a). No pathological changes were found in the other parts of the tracheobronchial tree. A fine-needle aspiration biopsy was taken. Having in mind systemic connective tissue disease at the beginning (particularly Wegener’s disease), specific antibody tests were performed (anti-ds DNA, pANCA, cANCA, C3, C4)—all in the reference range. An immunohistological examination with CD 138 was done—lymphoplasmic inflammatory infiltration, squamous metaplasia with dysplastic changes were found, more than 40% of lymphoplasmocytes marked with CD 138 express IgG4 (Figure 2a,b). Immunoproliferative hyper-IgG4 disease was diagnosed based on elevated serum IgG4, IgG4/IgG ratio > 0.4 and histological findings.

Corticosteroid treatment was initiated, at first systemic, after that orally, reaching a maintenance dose of 8 mg methylprednisolone daily. Later on, rigid bronchoscopy with partial resection of the stenotic fibers was performed, achieving considerable dyspnea reduction. During maintenance therapy three more control bronchoscopies were performed—a reduction of the stenotic ring and improved patency of the trachea were reported.

After two years of treatment (at the age of 17), the girl lacked subjective complaints—showed no shortness of breath or limitation of physical activity. The control spirometry was satisfactory: FVC—91%, FEV1—88%; normal blood acid-base state. The last fibrobronchoscopy showed a fibrous ring, through which a 4.4 mm fibrobronchoscope passed, followed by an unaffected trachea and annular cartilage. Dilation of the fibrous area was performed with rigid tube number 6.5 (Figure 1b).

As a result of long-term corticosteroid treatment (for a two-year period), the girl has iatrogenic Cushing—overrepresented subcutaneous adipose tissue, especially on the neck, weight gain of 8 kg, active stretch marks on the body, hypertrichosis.

Currently, oral steroid treatment in a maintenance dose is continued, no other immunosuppressive therapy has been resorted to. Close follow-up and the consideration of initiating biological treatment and/or surgical treatment are required in the case of relapse.

## 3. Discussion

To establish the diagnosis of this rare immunological disease as a cause of tracheal stenosis in a child, it is necessary, in addition to a positive immunological and histological diagnosis, to exclude, of course, other causes of tracheal stenosis. The most common causes of tracheal stenosis are prolonged intubation, tracheotomy or tracheostomy [7,8]. This leads to ischemia of the tracheal wall, ulceration and inflammation, damage to the cartilage rings, granulation tissue growth, scarring and the occurrence of narrowing of the tracheal lumen. Other causes of tracheal stenosis are foreign bodies, trauma to the neck and chest, inhalation damage, some diseases of the thyroid gland, thyroid tumors, lymph nodes infections, tracheitis caused by viruses and bacteria, including tuberculosis, autoimmune diseases (amyloidosis, sarcoidosis, Wegener’s granulomatosis), aortic aneurysm [9].

Our patient underwent prolonged therapy with various medications for bronchial asthma (she had a cough, easy fatigue during physical exertion, positive spirometry). The first bronchoscopy showed the stenotic area, a number of tests were performed to rule out other diseases such as tuberculosis, sarcoidosis, Wegener’s disease, and finally, after positive immunological tests (elevated IgG, IgG4 levels; IgG4/IgG ratio > 0.4) and histological tests, the diagnosis was definitely confirmed.

Immunoglobulin G4-related disease is an immune-mediated disease which could involve essentially any organ [10,11,12]. The epidemiology of this disease has not been completely explored. The majority of patients reported in the literature are from Japan, but the disease has been described all across the world. The mean age at diagnosis is approximately 60 years with male preponderance (male:female ratio is 8:3) [13].

The symptoms of this disease are determined by the affected organ. Generally, patients with IgG4-RD have few respiratory symptoms. Forty to fifty percent of patients with this disease have a history of allergic rhinitis and/or bronchial asthma and some have asthmatic symptoms, such as a cough and wheezing [14,15]. Airway manifestations may include rhinitis, sinusitis, asthmatic symptoms, and airway stenosis [16]. IgG4+ plasma cell infiltration was established in biopsies of the nasal mucosa of patients with rhino-sinusitis and biopsies of the bronchial mucosa of patients with asthmatic symptoms. These findings suggest a possible association between IgG4-RD and airway allergies [17].

Regarding therapy, in the active period, when a high serum concentration of IgG is established, there is a rapid clinical response to corticosteroid treatment [18,19]. The clinical experience is that most IgG4-RD patients respond highly to corticosteroid treatment, which induces remission in most patients [20]. A promising treatment in corticosteroid refractory patients is rituximab (RTX). RTX has been used in refractory IgG4-RD patients, the duration of treatment effect is not established [21,22,23]. Other immunosuppressive drugs like Mycophenolate mofetil, Azathioprine and Methotrexate are proposed by some authors in the treatment of IgG4-RD with moderate efficacy: about 50% remission rate [24]. Surgical solutions (laryngotracheal reconstruction surgery) should be chosen in cases of critical airway stenosis or in cases of no drug response [25].

Until now we have chosen to treat the girl only with corticosteroids with good response so far. The general condition as well as the respiratory function are regularly monitored.

Immunoglobulin G4-related disease is a relatively newly described disease and is sparsely identified in childhood and adolescence. One recent review article identified only 25 cases of childhood presentation up to 2016—none of them affecting the larynx or trachea [24]. Moreover, to the best of our knowledge, there is only one previous report of laryngeal involvement in childhood [26]. Therefore, we think that the current case may improve the understanding and increase familiarity of this generally unknown and multifaceted disease among pediatric practitioners.

## 4. Conclusions

The tracheal involvement of IgG4-RD is uncommon. Nonetheless, it is a manifestation that should be included in the differential diagnosis of the tracheal stenosis.

## Figures and Tables

**Figure 1 children-08-00589-f001:**
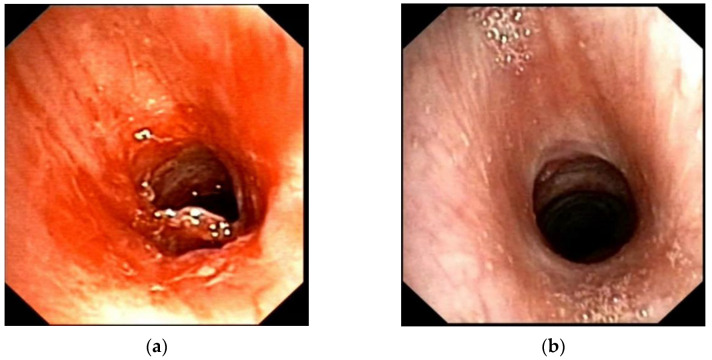
Endoscopic view. (**a**) Initial view—a fibrous stenosis at the level of the lower edge of the thyroid cartilage ring with an infiltration area; (**b**) a fibrous ring after dilation with rigid tube number 6.5, followed by unaffected trachea and annular cartilage.

**Figure 2 children-08-00589-f002:**
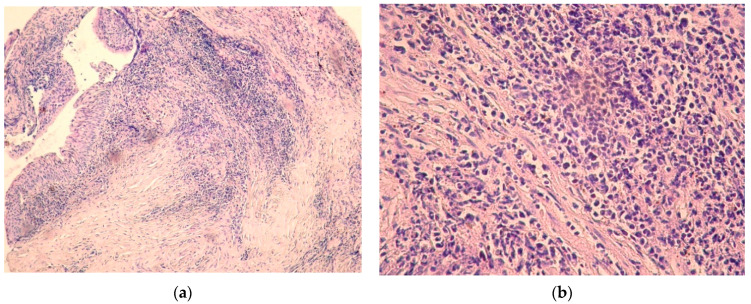
Histopathologic findings in tracheal biopsy specimen. (**a**) (Magnification 10 × 0.25) Diffuse lymphoplasmic inflammatory infiltration, squamous metaplasia with dysplastic changes by hematoxylin and eosin staining; (**b**) (magnification 40 × 0.65) plasma cell infiltration, specifically identified by CD 138 antibody. IgG4-positive plasma cells among the inflammatory infiltrates, showing a high IgG4/IgG ratio (>40%), by IgG4 immunostain.

## Data Availability

The data presented in this study are available on request from the corresponding author. The data are not publicly available due to privacy restrictions.

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
