# Peer review of "A Case of Tracheal Stenosis as an Isolated Form of Immunoproliferative Hyper-IgG4 Disease in a 17-Year-Old Girl"

_children, 2021, doi:10.3390/children8070589_

Round 1
Reviewer 1 Report
This manuscript is a case report of a young woman with tracheal stenosis caused by Hyper-IgG4-related disease. It describes the presentation, the diagnosis and the treatment of this young woman. In terms of a case report, it's novelty is that this is not an adult and the lesion is in the airway which is unusual.
I would like more detail in the presentation and the treatment. You say there is a positive bronchial dilation test on presentation, but no numbers are given. It would be useful to know how reactive she actually was and if PFT's were normal after bronchodilator (which goes against a completely fixed stenosis). Also no timeline is given--how long symptoms were there, were they progressive, etc. In terms of treatment--there is no timeline. How long did it take for this to resolve, how long to get to maintenance steroids. Also I do not know what was accomplished with the first "partial resection of the stenotic fibers." It would be useful to understand what had to be done to this young woman to improve her status.
There are some grammatical, punctuation, spelling issues and some terms are clearly not common in English. For instance, using "suffering" as the last word in the Introduction is not a real English term. You should say without intubation or other inciting event (or something similar). Gibbus in English usually refers to the kyphosis that occurs with spinal body collapse--is that really what you mean? Does this person have skeletal/bony issues due to the steroids? Please edit for more appropriate terms.
Author Response
Dear reviewer #1,
Тhank you for your constructive remarks and recommendations. In accordance to them, we have made the following corrections and additions to the manuscript (marked with their page and row in the revised manuscript – if applicable).
The result (value) from the bronchodilator test is added (2-56), which test is weakly positive and in itself is not confirmatory regarding the basic diagnosis of the girl. In early childhood, the patient had clinical symptoms of bronchial asthma, but therapeutic resistance led us to a more in-depth diagnostic testing. It is possible that the girl has both diseases, this is commented on in the discussion (4-124).
The symptoms - fatigue and persistent drug-resistant dyspnea are present from 3 years of age, without significant progression – present in the original manuscript (1- 44).
Тhe girl was under asthma therapy for 12 years, with no significant change in the presenting complaints – added in the text (2- 46).
After establishing the diagnosis, the patient was under two years of treatment and she lacked subjective complaints – showed no shortness of breath and limitation of physical activity (2-75).
With the first partial resection of the stenotic aria considerable dyspnea reduction was achieved (2-71).
In order to improve the girl's condition, we will continue regular follow-up, if necessary we will switch to additional immunosuppressive therapy, and surgical correction is being discussed (2- 84).
We made the corrections of the inappropriate terms, thank you for these recommendations.
Best regards
Natalia Gabrovska, MD
Reviewer 2 Report
This is very interesting especially for pediatricians. It is actually a very rare form of IgG4-RD. The paper describes very interesting and still not fully understood and classified entity of IgG4 - related disease. My concer refers to the final diagnosis of IgG4 - RD: is it definite or probable ? Please refer to the Boston criteria. What was the IgG4 concentrartion ? Other criteria of a definite diagnosis include the presence of IgG4 positive plasma cells in tissues in the IgG4/IgG positive cells ratio greater than 0,4 or IgG4 positive plasma cells of more than 10 per high power field. Treatment: In addition to steroids and rituximab another alternative is combination therapy. The combination of mycophenolate mofetil with azathioprine and methotrexate has been shown to be effective.
Author Response
Dear reviewer #2,
Тhank you for your constructive remarks and recommendations. In accordance to them, we have made the following corrections and additions to the manuscript (marked with their page and row in the revised manuscript – if applicable).
Yes, the diagnosis is definate based on high concentration of IgG, IgG4; IgG4 / IgG ratio> 0.4 (2-52), and positive histological result (2-63).
Yes, in terms of treatment, I added information about the available cytostatic drugs (4-130).
Best regards
Natalia Gabrovska, MD